# Functionalized Boron Nitride Nanosheets/Poly(l-lactide) Nanocomposites and Their Crystallization Behavior

**DOI:** 10.3390/polym11030440

**Published:** 2019-03-06

**Authors:** Deyu Kong, Deli Zhang, Hongge Guo, Jian Zhao, Zhaobo Wang, Haiqing Hu, Junting Xu, Cuiliu Fu

**Affiliations:** 1Key Laboratory of Rubber-Plastics Ministry of Education/Shandong Provincial Key Laboratory of Rubber-Plastics, Qingdao University of Science & Technology, No. 53 Zhengzhou Road, Qingdao 266042, China; and School of Materials Science and Engineering, Qilu University of Technology (Shandong Academy of Sciences), Jinan 250353, China; kdy0511@163.com (D.K.); zhangdl513@163.com (D.Z.); 921925576@qq.com (H.G.); wangzhb@qust.edu.cn (Z.W.); clfu@ciac.jl.cn (C.F.); 2State Key Laboratory of Molecular Engineering of Polymers, Fudan University, Shanghai 200433, China; 3Key Laboratory of Polymer Processing Engineering (South China University of Technology), Ministry of Education, Guangzhou 510640, China; 4MOE Key Laboratory of Macromolecular Synthesis and Functionalization, Department of Polymer Science and Engineering, Zhejiang University, Hangzhou 310027, China

**Keywords:** boron nitride, poly(l-lactide), nanocomposites, crystallization

## Abstract

In this work, hydroxyl-functionalized boron nitride nanosheet (OH-BNNS) was prepared and was blended with poly(l-lactide) (PLLA) to yield PLLA/OH-BNNS nanocomposites with excellent dispersion of OH-BNNS via the interaction of carbonyl in PLLA and hydroxyl in OH-BNNS. The effects of OH-BNNS on the crystallization and melting behaviors, isothermal crystallization kinetics, macroscopic crystal morphology and crystal structure of PLLA were studied by means of various techniques. The addition of OH-BNNS nanofillers can effectively accelerate the crystallization of PLLA and enhance the nucleation density, leading to a smaller spherulite size, increased crystallinity, a more obvious crystallization peak upon cooling but weakened cold crystallization behavior upon heating. Low OH-BNNS loading can increase the relative content of α-crystal, but the relative content of less perfect α′-crystal is increased at high OH-BNNS loading due to the strong interaction between PLLA and OH-BNNS.

## 1. Introduction

Boron nitride is a crystal composed of nitrogen atoms and boron atoms and contains four different variants, hexagonal boron nitride (h-BN), rhombohtic boron nitride (r-BN), cubic boron nitride (c-BN) and wurtzite boron nitride (w-BN), in which h-BN is very similar to graphite in structure, and commonly known as white graphite due to its own whiteness [1]. Over the past decade, few- and single- layered boron nitride nanosheets (BNNS) have attracted much attention due to their unique physical and chemical properties. BNNS is widely used in many fields, such as heat-resistant crucible, anti-oxidation lubricants, protective coatings and other barrier materials, dielectrics, catalysts and ceramics and so on [2,3,4]. The atoms in the h-BN layer are connected by σ bonds. The layers also have ionic bond characteristics in addition to van der Waals forces, that is, the “lip-lip” effect between adjacent layers, and the interaction between layers is strong. The stripping of BN is more difficult than that of graphene. The commonly used method of stripping bulk BN powder is direct solvent exfoliation assisted by sonication [5,6,7]. Functionalization is quite effective in overcoming the lip-lip interactions among the h-BN layers to achieve exfoliation. Several methods were used to prepare hydroxyl-modified BNNS (OH-BNNS) derivatives, including plasma treatment [8], NaOH-assistant ball-milling [9], treatment with H_2_O at high temperature [10], or using chemical reagents, which can generate OH radicals [11], etc.

As one of the focused research subjects in the field of bioplastics, poly(l-lactic acid) (PLLA) is a semi-crystalline thermoplastic resin with excellent biocompatibility and biodegradability [12,13], and its synthetic materials are renewable crops, such as corn, wheat, cassava and so on. Therefore, in the past decades, PLLA has attracted widespread attention as a kind of completely natural recyclable green polymer material [14,15,16,17,18,19]. Due to its thermoplasticity, high strength and high modulus, PLLA can be used in the industrial packaging field, producing molded parts or medical devices such as surgical sutures and drug delivery devices [20,21].

The chemical stability and mechanical properties of the crystalline polymer depend to a large extent on its crystal structure and crystalline morphology. Studying the crystallization behavior and crystal modification of PLLA is of guiding importance for the rational application of PLLA. Like most polyesters, PLLA usually has a low crystallization rate, and its crystal form and crystal structure vary widely upon heat treatment or solvent treatment [22]. It is found that PLLA has mainly four kinds of crystal structures, namely, α, β, γ, and ω crystal, corresponding to different helical chain conformations and arrangements of PLLA chains in a unit cell [23]. Among them, the α-crystal derives several sub-crystalline forms (α, α′, α′′ crystal forms). α-Crystal is the most well-known and also the most stable crystal form of PLLA, first proposed by DeSantis and Kovacs [24]. The forming conditions of α-crystal are relatively simple. It can be obtained from a melt sample, or a solution spinning at a low temperature and at a low drawing rate [24,25,26,27,28,29]. Zhang et al. [30] found that PLLA could form a more disordered new crystalline structure, named as α′-crystal, upon isothermal crystallization at lower temperatures (lower than 100 °C), which twists into 10/3 helical conformation due to the side CH_3_ interchain interactions. With an increase in crystallization temperature, the crystal form of PLLA begins to change into α-crystal [31].

Hexagonal boron nitride possesses many excellent physical and chemical properties such as outstanding chemical stability, high thermal conductivity, high electrical resistivity, excellent dielectric properties, good processability and mechanical strength. Therefore, boron nitride is commonly used as a reinforcing thermal-conductive filler for polymers [4,32,33,34,35]. Moreover, like other inorganic fillers [36,37,38,39,40,41], boron nitride can change the crystallization behavior of crystalline polymers. For example, boron nitride powder may act as nucleating agents to initiate the crystallization of poly(3-hydroxybutyrate) (PHB) and poly(3-hydroxybutyrate-*co*-3-hydroxyvalerate) (PHBV) [42]. To date, however, the effect of functionalized BNNS or exfoliated BNNS on crystallization behavior of polymers, especially PLLA, has not been reported in detail. It has been demonstrated that single- or few-layered graphene and graphene oxide (GO) can be well dispersed in the PLLA matrix to promote nucleation and accelerate crystallization of PLLA [36,37,38,39,40,41]. 

In our experiment, the ball milling method assisted by NaOH aqueous solution for modifying boron nitride nanosheets was mild, simple and widely applicable [9]. Hydroxyl functionalized boron nitride nanosheets (OH-BNNS) were produced by chemical milling and mechanical shearing in the simple ball milling process. Due to the interaction between hydroxyl groups in OH-BNNS and carbonyl groups of PLLA, OH-BNNS can be well dispersed in PLLA matrix. It was also found that OH-BNNS greatly accelerated PLLA crystallization and influenced the crystal structure and macroscopic crystal morphology of PLLA as well.

## 2. Experimental Section

### 2.1. Materials

Hexagonal boron nitride (B106033) was supplied by Aladdin. PLLA (3051D, *M*_w_ = 154 kDa, *M*_w_/*M*_n_ = 1.5) was purchased from Natureworks LLC (USA). *N*,*N*-dimethylformamide (DMF) (≥99.5%) was obtained from Tianjin Damao Chemical Reagent Factory. *n*-Hexane (≥99.5%) was provided by Tianjin BASF Chemical Co., Ltd. (Tianjin, China). Sodium hydroxide (NaOH) and hydrochloric acid (HCl) (37%) were purchased from Tianjin Reagents Co., Ltd. (Tianjin, China). These chemicals were used as received.

### 2.2. Preparation of Hydroxyl-Functionalized Boron Nitride Nanosheets (OH-BNNS)

2 g of boron nitride powder and 100 mL of NaOH aqueous solution (2M) were placed in a ball mill jar. The mill jar was placed on the ball mill. Ball milling time was set to 24 h and the speed was adjusted to 200 r/min. After the ball milling stopped, the sample was taken out, centrifuged, rinsed with HCl solution and filtered. Then the resulting sample was placed in a vacuum oven for drying. The hydroxyl-functionalized boron nitride (OH-BNNS) product was collected for follow-up characterizations. 

### 2.3. Preparation of PLLA/OH-BNNS Nanocomposites

10 g of PLLA was dissolved in a large amount of DMF and the concentration of the solution was less than 1 wt%. The mixture was stirred at 100 °C until PLLA was completely dissolved, and then the solution was sonicated for 0.5 h. 50 mg of OH-BNNS was added to 10 mL of DMF and the mixture was sonicated for 1 h. The suspension was added to the PLLA solution, followed by stirring. The solution containing the mixture of PLLA and OH-BNNS was heated to 100 °C to allow evaporation of most of the DMF. Subsequently, a large amount of *n*-hexane was added as a precipitation agent to the concentrated solution, followed by filtering. The product was dried in a vacuum oven at 60 °C for 48 h. The resulting sample was designated as PLLA-0.5. The same experimental procedure was used to prepare PLLA/OH-BNNS composites with OH-BNNS contents of 1.0, 2.0, and 3.0 wt% (labeled as PLLA-1, PLLA-2, and PLLA-3), respectively. The contents of OH-BNNS calculated from thermal gravity analysis (TGA) curves were 0.42, 0.90, 1.80 and 2.51 wt% for PLLA-0.5, PLLA-1, PLLA-2 and PLLA-3, respectively (see Appendix A). The sample without boron nitride but undergoing the same solvent treatment was labeled as PLLA-0, and the as-received PLLA was labeled as neat PLLA.

### 2.4. Characterizations

The thickness of OH-BNNS was evaluated using atomic force microscopy (AFM) (BRUKER NanoScope V MultiMode 8 instrument, Billerica, MA, USA) in tapping mode. The OH-BNNS suspension (in tetrahydrofuran) was spin-coated onto a silicon wafer. The microstructure of OH-BNNS was characterized by using transmission electron microscopy (TEM, JEOL 2010, Tokyo, Japan) at an accelerating voltage of 200 kV. A small amount of OH-BNNS was suspended in tetrahydrofuran (THF) and sonicated for a period of time. The samples for TEM were prepared by drying a droplet of the OH-BNNS suspension on a carbon-coated copper grid. X-ray diffraction (XRD) was performed on a D-MAX 2500/PC X-ray diffractometer (Rigaku Co., Tokyo, Japan), and CuK_α_ (λ = 0.15418 nm) was used as a radiation source. The current was 100 mA, and the voltage was 40 kV. The PLLA and PLLA/OH-BNNS nanocomposites were characterized after molding at 210 °C for 5 min, followed by cooling to room temperature at 10 °C·min^−1^. Polarized optical microscopy was used to examine the size and morphologies of PLLA crystals and was performed on an Olympus BX51 (Tokyo, Japan). Field-emission scanning electron microscopy (SEM) (Hitachi, S4800, Tokyo, Japan) was used to investigate the morphology of the PLLA nanocomposites at an acceleration voltage of 10 kV. The samples for SEM observation were prepared by dropping the dilute solution of PLLA/OH-BNNS nanocomposites in DMF on foil paper. Prior to SEM examination, the surfaces of samples were sprayed with Au. Infrared tests were carried out on a Bruker TENSOR27 Fourier infrared spectrometer. The OH-BNNS and h-BN samples for FTIR were prepared by tableting the thin films of neat PLLA or its nanocomposites prepared by solution dropping with KBr. The non-isothermal crystallization behavior and isothermal crystallization kinetics of the samples were investigated by DSC204F1 differential scanning calorimeter (Netzsch, Selb, Germany). The samples for testing the non-isothermal crystallization and melting behaviors were heated to 210 °C at 10 °C·min^−1^ under N_2_ atmosphere, held for 5 min to eliminate heat history, and then cooled to room temperature at a rate of 10 °C·min^−1^ and finally heating the sample again to 210 °C at 10 °C·min^−1^. The samples for isothermal cold crystallization were first cooled from the melt to room temperature at a rate of 50 °C·min^−1^ and then heated to the crystallization temperature for crystallization.

## 3. Results and Discussion

### 3.1. Characterization of OH-BNNS

In order to characterize the structural difference between the OH-BNNS and h-BN, the two dried samples were subjected to a FTIR test. As shown in Figure 1, OH-BNNS exhibited an obvious but broad peak at 3500 cm^−1^, which was attributed to the stretching vibration of hydroxyl groups (–OH). Both OH-BNNS and h-BN showed significant characteristic absorption peaks around 1360 cm^−1^ and 810 cm^−1^, corresponding to the in-plane and out-of-plane stretching vibrations of the B-N bond, respectively. One can see that, the intensity ratio of the peaks at 3500 cm^−1^ and 1360 cm^−1^ (*I*_3500_/*I*_1360_) in OH-BNNS was much larger than that in h-BN, indicating that BNNS was successfully functionalized with hydroxyl groups. The thickness and morphology of OH-BNNS were characterized by AFM, as shown in Figure 2. The sample for AFM analysis was prepared by deposition of an OH-BNNS suspension in THF onto the silicon substrate followed by drying. The thickness of the observed OH-BNNS was approximately 2.12 nm, indicating a few-layered structure. The lateral size of the OH-BNNS was approximately 250 nm. The morphology of OH-BNNS was also probed using transmission electron microscopy. As shown in Appendix A, ultrathin 2D nanoplatelets were observed. The structure of OH-BNNS was not damaged during the functionalization and exfoliation processes.

### 3.2. Macroscopic Crystal Morphology of PLLA/OH-BNNS Nanocomposites

The spherulites of polymers can be monitored by polarized optical microscopy. As shown in Figure 3, neat PLLA generated a smaller amount of spherulite but a larger spherulite size (~60 μm). The spherulite size gradually decreased and the number of spherulites gradually increased as the OH-BNNS content increased. That is, OH-BNNS was a nucleating agent for PLLA crystallization and the presence of OH-BNNS could accelerate crystallization of the polymer, leading to a smaller spherulite size but more spherulites. However, we also noticed that, at the OH-BNNS loading of 3.0 wt%, the spherulites became larger than those at 2.0 wt%, implying that the nucleation effect of OH-BNNS was retarded at high OH-BNNS content. The SEM observation (Appendix A) showed that the OH-BNNS nanoplatelets were well dispersed in the PLLA matrix.

### 3.3. Effect of OH-BNNS on Crystal Structure of PLLA

X-ray diffraction was used to reveal the crystal structure of PLLA/OH-BNNS nanocomposites. As shown in Figure 4, the PLLA/OH-BNNS nanocomposites showed four characteristic diffraction peaks around 2θ of 14.7, 16.6, 19.0 and 22.2°, corresponding to (010), (110)/(200), (203), and (015) planes of the α-crystal of PLLA, respectively [43]. Comparing the diffraction patterns of PLLA-0 and nanocomposites, one can see that the diffraction peak of the (002) crystal plane of functionalized boron nitride at 2θ = 27° [44,45] was hardly discerned, as compared with the noise signal. This implied that OH-BNNS was well dispersed in the PLLA matrix. PLLA can exhibit polymorphic crystal structures. For example, α′-crystal is frequently observed in PLLA with the co-existence of α-crystal. The α′-crystal of PLLA was slightly less ordered than the α-crystal, which exhibited characteristic diffraction peaks at 2θ = 16.5°, and 18.9°, respectively. We can see from Figure 4 that the 2θ values of four characteristic diffraction peaks were very similar for PLLA-0 and PLLA-0.5, PLLA-1 and PLLA-2. This meant that the addition of a small amount of OH-BNNS did not change the crystal structure of PLLA. However, it was noticed that the diffraction peaks of (110)/(200) and (203) crystal planes in PLLA-3 appeared at 2θ = 16.4°, and 18.8°, which were evidently smaller than those of α-crystal and were close to the values reported for α′-crystal. Such a shift of diffraction peaks to lower 2θ indicates that α′-crystal tends to form at high OH-BNNS loading.

Since the difference of PLLA α- and α′-crystals in diffraction position is quite small, we also used FTIR to characterize the crystal structure of PLLA and PLLA/OH-BNNS nanocomposites. The stretching vibration absorption peaks in the 1800–1700 cm^−1^ and 1250–1000 cm^−1^ regions were sensitive to the conformation, intrachain and interchain interactions of PLLA molecular chains, and thus the spectra in these two regions were mainly analyzed. Figure 5 and Figure 6 show the FTIR spectra of PLLA/OH-BNNS nanocomposites in the ranges of 1800–1700 cm^−1^ and 1250–1000 cm^−1^ and corresponding second derivative curves, respectively. The difference in the spectra between α- and α′-crystals can be clearly observed from the second derivatives. It can be seen from Figure 5 that, in addition to the main C=O stretching vibration peak at 1759 cm^−1^, three weak peaks were also split at 1749, 1768, and 1777 cm^−1^, which could not be observed in α′-crystal [31]. The stretching vibration peak of C–O–C bond and the rocking vibration peaks of –CH_3_ and –CH appeared in the range of 1250–1000 cm^−1^ (Figure 6), and the assignments of the infrared absorption peaks in this region are listed in Table 1. The two peaks at 1053 cm^−1^ and 1222 cm^−1^ were exclusively produced by the α-crystal of PLLA [31]. As a result, the intensity ratios of the peak at 1777 cm^−1^ over that at 1759 cm^−1^ (*I*_1777_/*I*_1759_) and the peak at 1222 cm^−1^ over that at 1213 cm^−1^ (*I*_1222_/*I*_1213_) were calculated and used to investigate the change of α- and α′-crystals with OH-BNNS loading (Figure 7). The larger the values of *I*_1777_/*I*_1759_ and *I*_1222_/*I*_1213_, the higher the relative content of α-crystal. It was noted that, when the OH-BNNS loading was lower, such as 0.5 and 1.0 wt%, the values of *I*_1777_/*I*_1759_ and *I*_1222_/*I*_1213_ became larger than those of PLLA-0, indicating a higher relative content of α-crystal. However, the values of *I*_1777_/*I*_1759_ and *I*_1222_/*I*_1213_ fell back at higher OH-BNNS loading. Particularly, the value of *I*_1222_/*I*_1213_ in PLLA-3 is lower than that of PLLA-0. This indicates that the relative content of α′-crystal in PLLA-3 was probably higher than that in PLLA-0, in good agreement with the XRD results. The change of the relative content of α- and α′-crystals with OH-BNNS loading may be related to the hydrogen (H)-bonding interaction between OH-BNNS and PLLA. The hydroxyl groups in OH-BNNS can form H-bonds with the carbonyl groups in PLLA, which was evidenced by the slight shift of the main vibration peak of C=O group to a low wavenumber with the increasing loading of OH-BNNS (Figure 5). The moderate H-bonding interaction between OH-BNNS and PLLA at low OH-BNNS loading can induce the partially ordered arrangement of PLLA chains in the melt, which is favorable to the subsequent crystallization and formation of more ordered α-crystal. However, at high OH-BNNS loading, the H-bonding interaction between OH-BNNS and PLLA may be too strong, and the restriction effect of OH-BNNS may become more pronounced due to the shorter distance between two adjacent OH-BNNS platelets [46]. Both factors will lead to lower mobility of PLLA chains and formation of more α′-crystals, which is less ordered. 

### 3.4. Crystallization and Melting Behaviors of PLLA/OH-BNNS Nanocomposites

The crystallization and melting behaviors of PLLA and PLLA/OH-BNNS nanocomposites were investigated by DSC, and the obtained glass transition temperature (*T*_g_), onset and end temperatures of glass transition (*T*_g_^onset^ and *T*_g_^end^), crystallization peak temperature (*T*_c_) in the cooling process, and cold-crystallization peak temperature (*T*_cc_), melting peak temperature (*T*_m_) and melting enthalpy (Δ*H*_f_) in the second-run heating process are summarized in Table 2. Crystallinity *X*_c_ can be determined using the following formula:(1)$$X_{c}\;=\;\frac{\Delta H_{f}}{\left(1\;-\;\varphi \right)\Delta H^{*}}\;\times \;100\%$$
where Δ*H*_f_ was the measured melting enthalpy of samples, Δ*H** was the melting enthalpy of the PLLA with 100% crystallinity (93 J/g), and *φ* was the mass fraction of OH-BNNS in the PLLA/OH-BNNS nanocomposites [12].

The DSC cooling and second-run heating curves of the PLLA/OH-BNNS nanocomposites are illustrated in Figure 8. Firstly, the samples of PLLA-0, PLLA-0.5, PLLA-1 and PLLA-2 had similar *T*_g_, but the *T*_g_ of PLLA-3 was apparently higher than those of the other four samples (Table 2), which may be ascribed to the stronger H-bonding interaction between PLLA and OH-BNNS in PLLA-3 and a more pronounced restriction effect of OH-BNNS on PLLA chains. The *T*_g_^onset^ varied irregularly with the content of OH-BNNS, but all the PLLA/OH-BNNS nanocomposites exhibited a higher *T*_g_^onset^ than the neat PLLA. On the other hand, the *T*_g_^end^ gradually increased as the content of OH-BNNS increased. No obvious crystallization peak was observed in the cooling traces of PLLA-0 due to a very slow crystallization rate, while PLLA-0.5 and PLLA-1 samples exhibited a broad and weak crystallization peak centered at 98.3 °C and 100.1 °C, respectively. PLLA-2 and PLLA-3 exhibited a strong exothermic peak upon cooling from the melt (Figure 8a). However, the *T*_c_ of PLLA-3 was lower than that of PLLA-2, indicating weaker crystallizability of PLLA-3. This is in accordance with the POM, XRD and FTIR analysis. On the other hand, PLLA-0, PLLA-0.5 and PLLA-1 exhibited a broad cold-crystallization peak prior to melting in their second-run melting traces. The *T*_cc_ decreased gradually with the increasing OH-BNNS loading. Usually a lower *T*_cc_ indicated stronger crystallizability. Therefore, addition of a small amount of OH-BNNS can enhance the crystallizability of PLLA, as supported by the POM result. No cold-crystallization peak was observed for PLLA-2 and PLLA-3, showing that both samples had crystallized completely in the previous cooling process. All the samples exhibited a broad or overlapped melting peak, reflecting the complicated melting behavior due to co-existence of α- and α′-crystals as well as cold-crystallization [47]. The *T*_m_s of PLLA/OH-BNNS nanocomposites fluctuated slightly with varied OH-BNNS loading because of the broad melting peaks. Nevertheless, we noticed that the *T*_m_ of PLLA-3 was obviously lower than the other four samples, probably due to a higher content of less ordered α′-crystal. Moreover, the melting enthalpy and crystallinity also increased as the OH-BNNS loading increased, confirming that the presence of OH-BNNS can promote the crystallizability of PLLA. However, the crystallinity of PLLA-3 was lower than that of PLLA-2, revealing that too high OH-BNNS loading is unfavorable to PLLA crystallization.

### 3.5. Isothermal Crystallization Kinetics of PLLA/OH-BNNS Nanocomposites

The isothermal cold crystallization of PLLA and PLLA/OH-BNNs nanocomposites was carried out at various temperatures ranging from 116 to 132 °C. The heat flow curves of PLLA-0 and PLLA-2 during isothermal cold crystallization are shown in Figure 9 and the heat flow curves of other PLLA/OH-BNNS nanocomposites are given in the Appendix A. Since the samples were already cooled to room temperature before isothermal cold crystallization, in which crystal nuclei had been generated, the isothermal cold crystallization rate mainly depended on the number of pre-formed nuclei [48]. One can see from Figure 9 that the heat flow peak shifted to a longer crystallization time as the isothermal cold crystallization annealing temperature *T*_a_ increased, showing the reduced crystallization rate at higher *T*_a_. The observation was similar to that reported in literature [49]. Moreover, PLLA-2 finished its crystallization much earlier than PLLA-0 at the same *T*_a_, indicating a faster crystallization rate due to the incorporation of OH-BNNS. 

The relative crystallinity (*X*_t_) was proportional to the heat released during crystallization and is described by the following formula:(2)$$X_{t}=\frac{\int_{0}^{t}\left(dH/dt\right)dt}{\int_{0}^{\infty }\left(dH/dt\right)dt}$$

The numerator represents the heat generated from the beginning of crystallization (*t* = 0) to crystallization time *t*, and the denominator represents the total heat generated during the entire crystallization process. 

Figure 10 shows the variation of relative crystallinity with crystallization time for PLLA-0 and PLLA-2. From the *X*_t_ − *t* curves, one can also obtain the crystallization half-time (*t*_1/2_), which is defined as the crystallization time at which *X*_t_ reaches 50%. Crystallization half-time is one of the most important characteristic parameters in crystallization kinetics, and is commonly used to evaluate the crystallization rate of polymers. A larger *t*_1/2_ means a slow crystallization rate.

The isothermal crystallization kinetics of polymers can also be analyzed with the Avrami equation:1 − *X*_t_ = exp (−*kt^n^*)(3)
where *n* is the Avrami exponent and k is the crystallization rate constant. The value of *n*, usually ranging from 1.0 and 4.0, depends on the nucleation mechanism and dimension of the crystal growth.

Equation (3) can be re-formed as follows: ln[−ln(1 − *X*_t_)] = *n* ln*t* + ln *k*(4)

A linear relationship can be obtained by plotting ln[−ln(1 − *X*_t_)] versus ln*t*. The slope is the Avrami exponent *n* and the intercept is ln*k*. Figure 11 shows the Avrami plots for PLLA-0 and PLLA-2, and the obtained data of *n* and *t*_1/2_ are summarized in Table 3.

It can be seen from Table 3 that the content of OH-BNNS in the PLLA/OH-BNNS nanocomposites had no significant effect on the value of *n*. Since the crystal nuclei were pre-formed in the cooling process, the value of *n* was mainly related to the growth dimension of crystals. This showed that the addition of OH-BNNS did not alter the growth dimension of PLLA crystals. The values of *t*_1/2_ at various *T*_a_s are plotted in Figure 12 and some selected values are listed in Table 3. The data in Figure 12 reveal that the *t*_1/2_ increased with the increasing *T*_a_, indicating greatly retarded crystallization at higher *T*_a_. On the other hand, *t*_1/2_ basically became smaller as OH-BNNS loading increased, showing that the addition of OH-BNNS can accelerate PLLA crystallization, especially at higher *T*_a_s. This can be attributed to more heterogeneous nuclei formed by OH-BNNS at higher loading of OH-BNNS. However, we noticed that PLLA-0.5 exhibited values of *t*_1/2_ even larger than PLLA-0 at higher *T*_a_s. This could be explained by the fact that two crystallization processes may occur in PLLA-0.5 with a low OH-BNNS loading [50]. The PLLA chains contacting OH-BNNS crystallized faster due to heterogeneous nucleation, while the PLLA chains far from OH-BNNS crystallized at a much slower rate and may have been spatially trapped into and/or among the crystals formed via heterogeneous nucleation. The entrapped PLLA chains even crystallized more slowly than the unconfined simple PLLA, leading to a larger *t*_1/2_ than PLLA-0.

## 4. Conclusions

The hydroxyl group of OH-BNNS forms a hydrogen-bond with the carbonyl group of PLLA, contributing to improved dispersion of OH-BNNS in the polymer matrix. The polarizing microscope showed that a small amount of OH-BNNS could nucleate PLLA crystallization, forming small-sized spherulites. XRD analysis confirmed that PLLA/OH-BNNS nanocomposites formed mixed α-crystal and α′-crystal. FTIR results revealed that the relative content of α-crystal was higher at low OH-BNNS loading. However, the relative content of less perfect α′-crystal was increased at high OH-BNNS loading, which may have been attributed to too strong interaction between PLLA and OH-BNNS. OH-BNNS can also accelerate crystallization and enhance crystallinity of PLLA, indicative of a nucleating agent effect of OH-BNNS. The crystallization mechanism and crystal structure remained unchanged for both neat PLLA and PLLA/OH-BNNS nanocomposites.

## Figures and Tables

**Figure 1 polymers-11-00440-f001:**
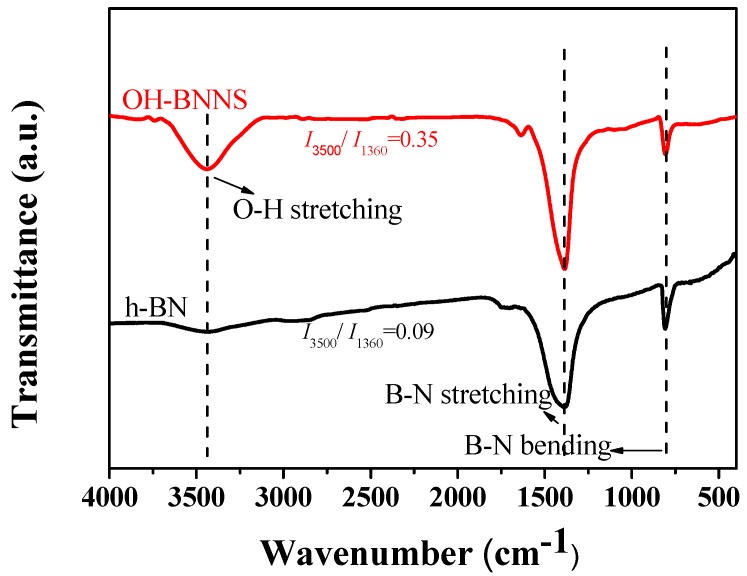
The FTIR spectra of h-BN and OH-BNNS. The *Y* axis was shifted for clarity.

**Figure 2 polymers-11-00440-f002:**
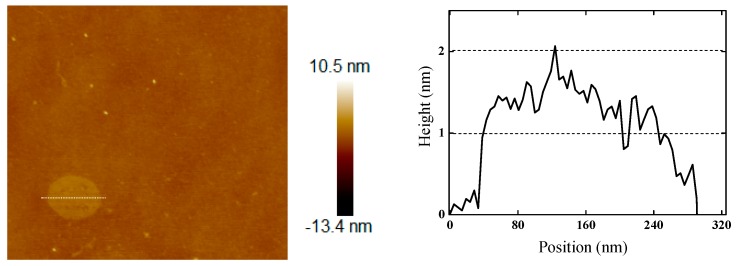
AFM topography image of few-layered OH-BNNS and the corresponding height profile.

**Figure 3 polymers-11-00440-f003:**
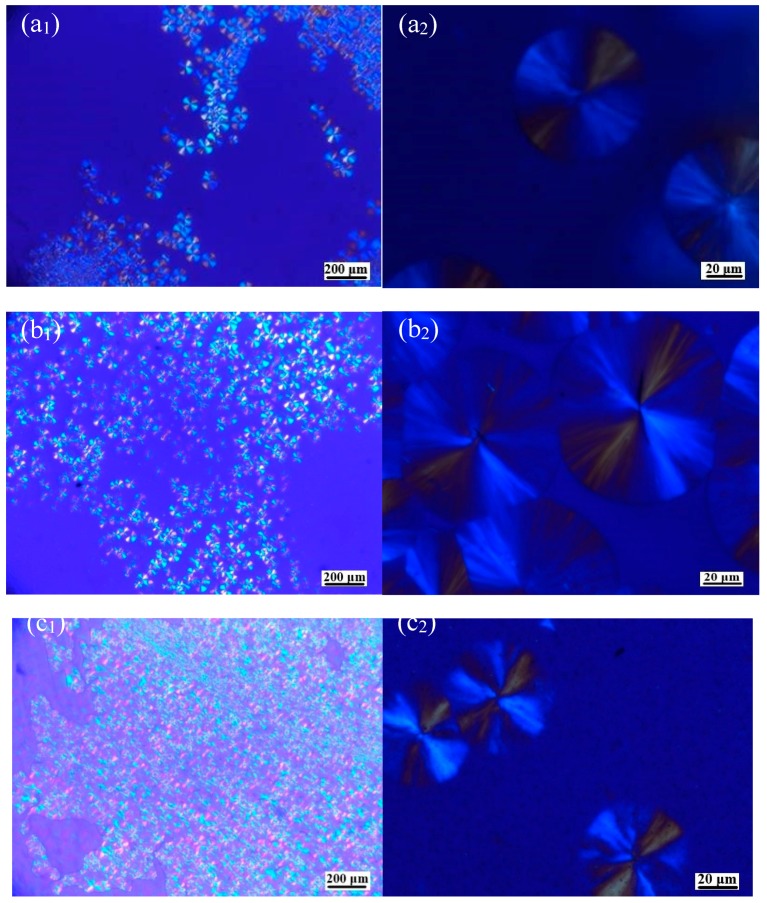
Polarized optical micrographs of PLLA containing different amounts of BNNS after being cooled from the melt (**a1**,**a2**,**b1**,**b2**,**c1**,**c2**,**d1**,**d2**,**e1**,**e2**,**f1**,**f2** are for neat PLLA, PLLA-0, PLLA-0.5, PLLA-1, PLLA-2, and PLLA-3, respectively, with different magnifications).

**Figure 4 polymers-11-00440-f004:**
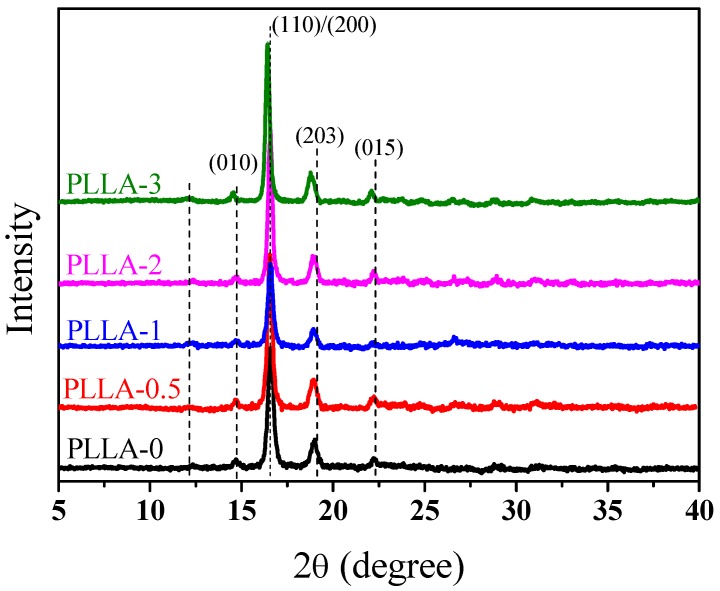
WAXD patterns of PLLA and PLLA/OH-BNNS nanocomposites with different OH-BNNS contents.

**Figure 5 polymers-11-00440-f005:**
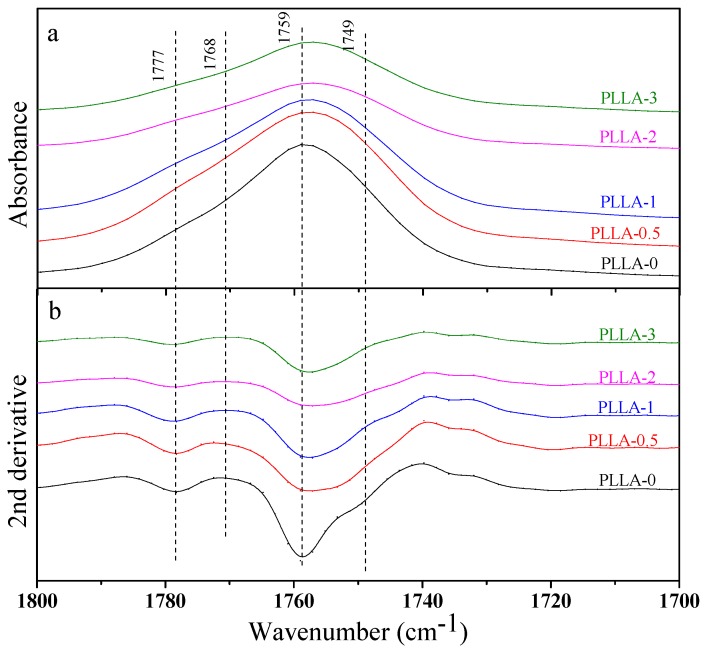
The FTIR spectra (**a**) and corresponding second derivatives (**b**) of PLLA and PLLA/OH-BNNS nanocomposites in the range of 1800–1700 cm^−1^.

**Figure 6 polymers-11-00440-f006:**
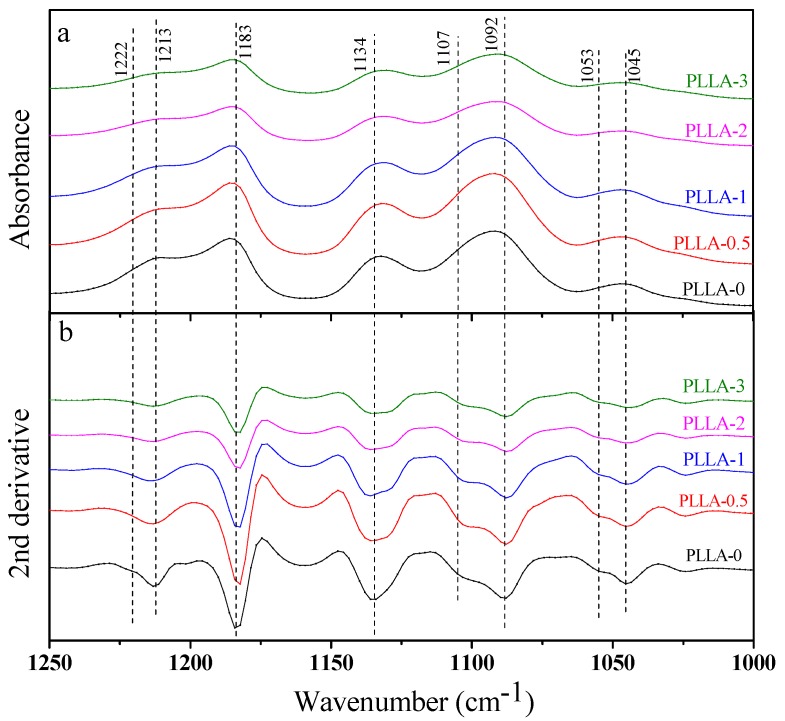
The FTIR spectra (**a**) and corresponding second derivative curves (**b**) of PLLA/OH-BNNS nanocomposites in the range of 1250–1000 cm^−1^.

**Figure 7 polymers-11-00440-f007:**
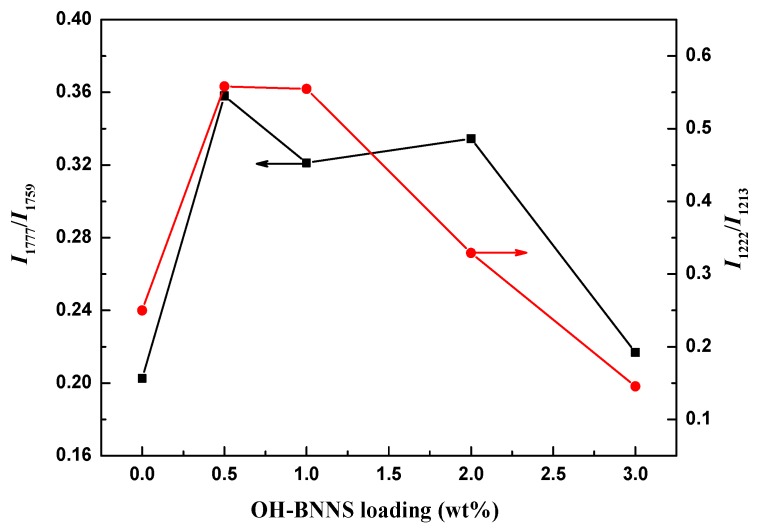
Intensity ratios of the peaks at 1749 cm^−1^ to 1759 cm^−1^ and the peaks at 1222 cm^−1^ to 1213 cm^−1^ in the second derivative curves.

**Figure 8 polymers-11-00440-f008:**
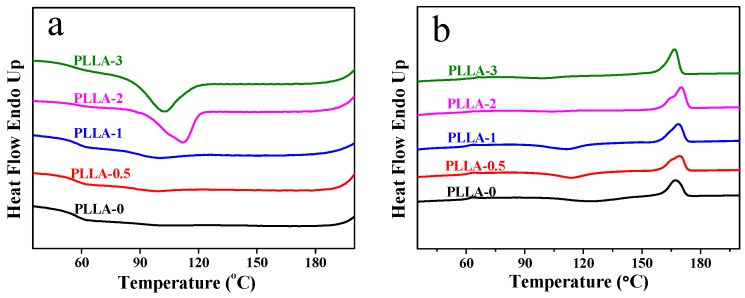
(**a**) DSC cooling and (**b**) second-run heating curves of PLLA/OH-BNNS nanocomposites.

**Figure 9 polymers-11-00440-f009:**
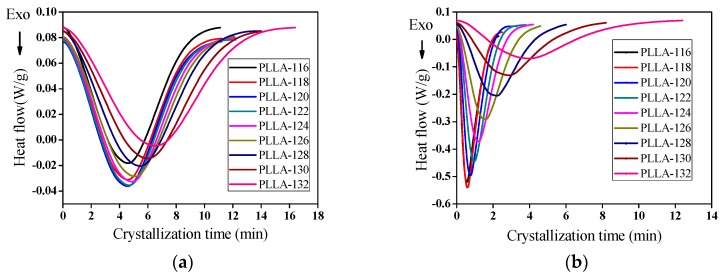
Heat flow curves of (**a**) PLLA-0 and (**b**) PLLA-2 during isothermal cold crystallization at different crystallization temperatures.

**Figure 10 polymers-11-00440-f010:**
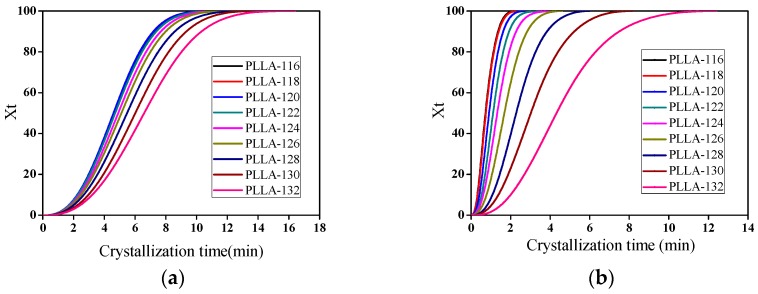
Evolution of relative crystallinity with crystallization time for PLLA-0 (**a**) and PLLA-2 (**b**).

**Figure 11 polymers-11-00440-f011:**
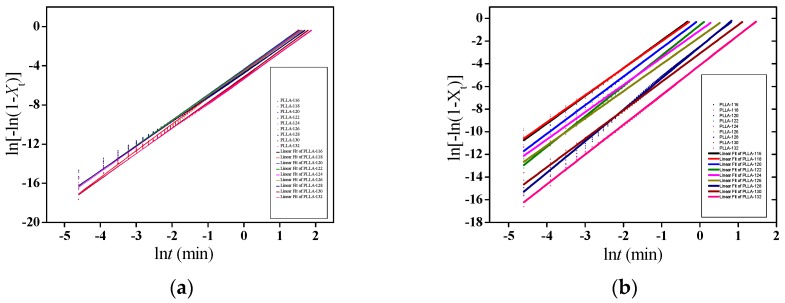
Avrami plots for PLLA-0 (**a**) and PLLA-2 (**b**).

**Figure 12 polymers-11-00440-f012:**
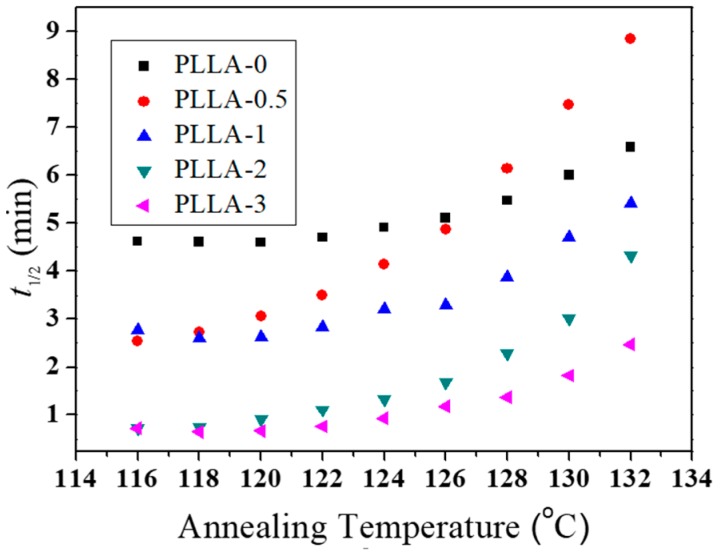
*t*_1/2_ as a function of *T*_a_ for PLLA/OH-BNNS nanocomposites.

**Table 1 polymers-11-00440-t001:** Assignments for the FTIR bands in the 1250–1000 cm^−1^ region for PLLA α′ and α crystals [31].

IR Frequencies (cm^−1^)		Assignments
α′	α	
1213	1213	ν_as_(C–O–C) + r_as_(CH_3_)
	1222	
1183	1183	
1134	1134	r_s_(CH_3_)
1092	1092	ν_s_(C–O–C)
1107	1107	
1045	1045	ν(C-CH_3_)
	1053	

**Table 2 polymers-11-00440-t002:** The results from the DSC curves for PLLA/OH-BNNS nanocomposites.

Sample	*T*_g_ (°C)	*T*_g_^onset^ (°C)	*T*_g_^end^ (°C)	*T*_c_ (°C)	*T*_cc_ (°C)	*T*_m_ (°C)	Δ*H*_f_ (J/g)	*X*_c_ (%)
PLLA-0	60.2	41.1	65.1	-	123.7	167.2	35.2	37.9
PLLA-0.5	60.5	43.5	65.5	98.3	114.1	169.3	35.4	38.3
PLLA-1	60.4	42.7	66.3	100.1	111.3	168.7	38.2	41.5
PLLA-2	60.6	44.4	67.5	112.1	-	170.1	44.4	48.7
PLLA-3	64.9	45.1	70.3	102.5	-	166.7	40.4	44.8

**Table 3 polymers-11-00440-t003:** Isothermal crystallization kinetics results of PLLA/OH-BNNS nanocomposites.

BNNS Content (wt%)	*T*_a_ (°C)	*t*_1/2_ (min)	*n*
0	122	4.70	2.58
0	128	5.48	2.50
0.5	122	3.50	2.59
0.5	128	6.15	2.61
1	122	2.84	2.63
1	128	3.87	2.52
2	122	1.11	2.68
2	128	2.28	2.78
3	122	0.76	2.41
3	128	1.37	2.19

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
