# Peer review of "Functionalized Boron Nitride Nanosheets/Poly(l-lactide) Nanocomposites and Their Crystallization Behavior"

_polymers, 2019, doi:10.3390/polym11030440_

Round 1

Reviewer 1 Report

The manuscript is devoted to the influence of hydroxyl derivative of boron nitride BN on the crystallization behavior of poly(lactid acid), PLLA -  this is the aim of the work. The Authors use several experimental techniques to follow the PLLA crystallization at various MN loadings. In general, the work was done and written correctly. However, I have a few comments; I would ask the authors to turn their attention to them in the revised version.

1.       I would like to know the possible use of such a composite, because the main reason for the application of PLA is its biodegradability, and BN is not biodegradable.

2.       Fig. 3. The segment length mark is very difficult to read

3.       p. 6: “The SEM observation (Figure S3) shows…” It should be Fig. S2.

4.       p. 8. Fig. 5.Why deconvolution has not been made?

5.       p. 8. “The hydroxyl groups in OH-BNNS can form H-bonds with the carbonyl groups in PLLA, which is evidenced by the slight shift of the main vibration peak of C=O group to low wavenumber with the increasing loading of OH-BNNS (Figure 5).” I have some doubt concerning this statement. The number of C=O groups is probably much higher than that of OH groups, so only a part of carbonyl groups will be H-bonded. In such a case we would observe a shoulder on the main C=O peak, not the shift of the whole peak.

6.       p. 9. “However, the H-bonding interaction between OH-BNNS and PLLA may become too strong at high OH-BNNS loading…” This is a composite and I think that not only H-bonding affects its properties.

Reviewer 2 Report

The paper titled "Functionalized Bron Nitride Nanosheets/PLLA Nanocomposites and Their Crystallization Behavior" deals with the surface modification of BN nanoparticles, in order to improve their compatibilty with PLLA, and with the following blend of PH-BNNS nanoparticles with PLLA via solution blending.

The paper is interesting and well written, therefore I recommend publication after some minor revisions

In 2.1 Materials, authors do not indicate the purity of solvents and reagents

In line 103-104, I don't understand what do authors mean with the term "concentrated": PLLA is dissolved in DMF, then stirred and OH-BNNS dispersed in DMF is later added. No concentration procedure is described

I suggest authors to check the real content of PH-BNNS in the nanocomposites via TGA: it can happen that either real concentration is lower or higher due to some residues remaining on the filters (Authors should indicate pore size of the filters) or to some differences in the local concentration of BN dispersed in PLLA, given the very different densities of BN and PLLA respectively

In Figure 1, the intensity ratio is described: it does not seem reliable to me, given the differences in the baseline. It is anyway evident that -OH signals are present in OH-BNNS

Line 181: what do authors mean by "not obvious"?

In DSC curves, authors talk about Tg values, but in PLLA-2 and PLLA-3 Tg is not well visible and, most important, the temperature range in which Tg occurs is higher than in the other samples. Therefore I do not think that the Tg value alone can give useful information: also the onset and endset of the transition could be interesting

Make some check on English: some minor mistakes are present and some commas should be removed

Reviewer 3 Report

Well written paper with a good experimental approach. My recommendation is to publish as such.
